# Accuracy of Anti-SARS-CoV-2 Antibody in Comparison with Surrogate Viral Neutralization Test in Persons Living with HIV, Systemic Lupus Erythematosus, and Chronic Kidney Disease

**DOI:** 10.3390/vaccines12050558

**Published:** 2024-05-20

**Authors:** Marita Restie Tiara, Chrisan Bimo Prayuda, Tara Titian Maulidya, Hofiya Djauhari, Dadang Suhendar, Rudi Wisaksana, Laniyati Hamijoyo, Rudi Supriyadi, Agnes Rengga Indrati, Bachti Alisjahbana

**Affiliations:** 1Research Center for Care and Control of Infectious Disease, Universitas Padjadjaran, Bandung 40161, Indonesia; marita.restie@gmail.com (M.R.T.); chrisan.bp@gmail.com (C.B.P.); taratitian19@gmail.com (T.T.M.); hofiya@gmail.com (H.D.); rudi.wisaksana@unpad.ac.id (R.W.); agnes.indrati@unpad.ac.id (A.R.I.); 2Mayapada Hospital Buah Batu, Bandung 40266, Indonesia; 3Research Center for Applied Microbiology, National Research and Innovation Agency (BRIN), KST Soekarno, Cibinong, Jl. Raya Jakarta—Bogor KM 46, Cibinong 16911, Indonesia; dadang.suhendar@brin.go.id; 4Infectious and Tropical Disease Division, Internal Medicine Department, Hasan Sadikin Hospital, Faculty of Medicine, Universitas Padjadjaran, Bandung 40261, Indonesia; 5Rheumatology Division, Department of Internal Medicine, Hasan Sadikin Hospital, Faculty of Medicine, Universitas Padjadjaran, Bandung 40161, Indonesia; laniyati.hamijoyo@unpad.ac.id; 6Nephrology and Hypertension Division, Internal Medicine Department, Hasan Sadikin Hospital, Faculty of Medicine, Universitas Padjadjaran, Bandung 40161, Indonesia; rudi.supriyadi@unpad.ac.id; 7Clinical Pathology Department, Hasan Sadikin Hospital, Faculty of Medicine, Universitas Padjadjaran, Bandung 40161, Indonesia

**Keywords:** COVID-19, vaccination, antibody, immunoassay

## Abstract

The presence of the anti-SARS-CoV-2-RBD antibody (anti-RBD) prevents severe COVID-19. We aimed to determine the accuracy of a point-of-care anti-RBD testing implemented in persons living with HIV (PLWH), systemic lupus erythematosus (SLE), and chronic kidney disease (CKD). We enrolled 182 non-comorbid subjects and 335 comorbid subjects (PLWH, SLE, CKD) to test the anti-RBD assay compared to the surrogate viral neutralization test (sVNT) as the reference test. We performed linear correlation analysis between anti-RBD and sVNT, along with an ROC analysis to ascertain the anti-RBD cutoff at 30%, 60%, and 90% inhibition of sVNT, to calculate accuracy. The correlations between anti-RBD and sVNT among all groups were excellent, with R = 0.7903, R = 0.7843, and R = 0.8153 among the non-comorbid, SLE, and CKD groups, respectively, and with significantly higher correlation among the PLWH group (R = 0.8877; *p*-value = 0.0072) compared to the non-comorbid group. The accuracy of the anti-RBD test among the PLWH and CKD groups was similar to that among the non-comorbid group but showed lower sensitivity in the SLE group (*p* = 0.000014). The specificity of the test remained high in all groups. In conclusion, the anti-RBD test had excellent correlation with the sVNT. The persistently high specificity in all groups suggests that this test can be reliably utilized to detect the presence of low neutralization capacity, prompting additional vaccination.

## 1. Introduction

Coronavirus disease 2019 (COVID-19) is a disease that infects the respiratory system, caused by severe acute respiratory syndrome coronavirus 2 (SARS-CoV-2). The virus can be transmitted directly from person to person by aerosols and small droplets, or indirectly by contact with contaminated surfaces [1]. Thus, it spreads widely and is highly contagious. Coronavirus disease 2019 (COVID-19) was declared a global pandemic by the World Health Organization (WHO) in March 2020, contributing to increased mortality rates and impacting various sectors worldwide [2].

Vaccinations against COVID-19 have proven to be successful in preventing serious illness. According to the policies implemented by the WHO and health ministries, everyone must receive vaccination to protect themselves against severe symptoms and minimize the risk of transmission [3]. The current situation shows that COVID-19 vaccines have contributed significantly to controlling the COVID-19 pandemic [4]. On the other hand, several reports have shown that the efficacy of COVID-19 vaccination is lower in patients with comorbidities than in healthy individuals. Patients with comorbidities may have reduced humoral and cellular immune responses [5].

The plaque reduction neutralization test is the best assay to measure the level of immune response. However, this test is challenging and can only be performed in a well-resourced laboratory. The surrogate viral neutralization test (sVNT) is a more accessible alternative that can be conducted in standard immunological labs and has been well received as a reference [6]. The measurement of antibody levels using the anti SARS-CoV-2-S-RBD antibody test is another examination that is regarded as being simple to perform. The fluorescence-based rapid anti-S-RBD test can even be utilized in a field laboratory. Several reports have shown the excellent accuracy of this test [7]. However, due to the known effect of comorbidities being associated with a lower immune response, we wanted to know whether the presence of comorbid conditions affects the sVNT’s accuracy performance [8].

This study aimed to determine the accuracy of the anti-SARS-CoV-2-RBD antibody test compared to the surrogate viral neutralization test among persons living with HIV (PLWH), systemic lupus erythematosus (SLE), and chronic kidney disease (CKD).

## 2. Materials and Methods

### 2.1. Study Design

We conducted a diagnostic study comparing the accuracy of a point-of-care anti-SARS-CoV-2-RBD antibody test (FastBio-RBD) compared to the GenScript sVNT as the reference standard among subjects with comorbidities. We enrolled subjects with no comorbidities, PLWH, SLE, and CKD in stable condition. The PLWH group involved subjects who were diagnosed with HIV and had received anti-retroviral therapy at the HIV Clinic at Hasan Sadikin General Hospital. Subjects with SLE were registered patients who had already been diagnosed with SLE and received routine follow-ups at the rheumatology clinic at Hasan Sadikin General Hospital. The CKD group comprised end-stage renal disease patients on hemodialysis at Hasan Sadikin General Hospital and Slamet General Hospital, Garut, West Java, Indonesia. The enrollment of each group was conducted at different timepoints. The non-comorbid subjects were enrolled from April 2021 to August 2022. The PLWH group was enrolled on September 2021, the SLE group from November 2021 to February 2022, and the CKD group from September to December 2021. All subjects were enrolled at a single timepoint and consecutively, based on their visit to the clinic at their convenience.

### 2.2. Data Collected

Data collected regarding subject characteristics included sex, age, and current comorbidities. We also collected data regarding history of previous COVID-19 infection and vaccination status. We defined previous COVID-19 infection as having had respiratory symptoms with a positive PCR result or a positive antibody test in the past. This was only assigned when COVID-19 was documented as no longer current. Subjects who had received COVID-19 vaccines were classified into several categories: unvaccinated, CoronaVac only, ChAdOx1-S only, or mRNA (BNT162b2 or mRNA-1273) vaccines. Subjects who had received any number of CoronaVac vaccines with subsequent mRNA boosters were classified in the mRNA group. CoronaVac is a whole-virus inactivated vaccine developed by Sinovac Biotech, Beijing, China. It was the first vaccine delivered in the Indonesian population, used since March 2021.

### 2.3. Anti-RBD: The Point-of-Care Anti-SARS-CoV-2-RBD Antibody Test

We used the anti-RBD test produced by Wondfo Biotech Co., Ltd. (Guangzhou, China), branded as the FastBio-RBD test for distribution in Indonesia by PT Biofarma Indonesia, Bandung, Indonesia (Persero). The test was carried out in accordance with the manufacturer’s instructions [9]. The FastBio-RBD test is a point-of-care test that detects the SARS-CoV-2 RBD antibody based on fluorescence immunoassay principles. This test detects total antibodies against the SARS-CoV-2-S-RBD antigen. The platform relies on a sandwich reaction, where the wild-type S-RBD antigen is present in the test line. After being combined with the detection buffer, the serum sample is added to the sample well. Serum anti-S-RBD antibodies from the patient will bind to the RBD antigen, which is coupled with a phosphorescent marker to generate immune complexes. When the serum–buffer mix is applied to the kit, it will migrate onto the nitrocellulose membrane and be caught by other RBD antigens on the test line. The associated fluorescence immunoassay (FIA) meter is then used to identify the resultant complex. The intensity of the fluorescence obtained from the immunochromatographic test allows for quantification. Anti-RBD levels were expressed in arbitrary units (AU/mL), with 0 and 200 AU/mL represent the lowest and greatest concentrations, respectively. A measurement above 1 AU/mL was considered positive [9,10].

### 2.4. sVNT: GenScript cPass SARS-CoV-2 Neutralization Antibody Detection Kit

The surrogate viral neutralization test (sVNT) was measured using the GenScript cPass SARS-CoV-2 Neutralization Antibody Detection Kit (GenScript Biotech, Leiden, the Netherlands). The test was performed according to the manufacturer’s instructions [11]. The GenScript SARS-CoV-2 sVNT kit detects the presence of neutralizing antibodies against SARS-CoV-2 circulating in human serum or plasma by a competitive mechanism. The presence of neutralizing antibodies against SARS-CoV-2 blocks the interaction between the receptor-binding domain (RBD) of the viral spike glycoprotein and the ACE2 cell surface receptor available within the reaction. The degree of inhibition, as measured by enzyme-linked serological assay, was recognized as the level of neutralizing antibody. The kit contains two components: the horseradish peroxidase (HRP)-conjugated recombinant SARS-CoV-2 RBD fragment (HRP-RBD), and the human ACE2 receptor protein (hACE2). The RBD in this kit originates from the Beta variant of SARS-CoV-2 (B.1.351, South Africa), which contains three mutations: K417 N, E484K, and N501Y [12]. The antibody level is measured by % inhibition, ranging from minimum to maximum inhibition of 0 to 100%, respectively. A result of >30% inhibition was regarded as positive.

### 2.5. Statistical Analysis

We used frequency distribution tabulation to describe the following subject characteristics: age, gender, comorbidities, history of COVID-19 infection, and vaccination status. We also described the distribution of the anti-RBD and % inhibition in the table and plotted their values, stratified based on comorbidity groups and vaccination status.

We plotted and analyzed the results of the anti-RBD titers measured with the anti-RBD AU value against the % inhibition measured with the sVNT using Spearman’s ranked correlation. The R values were compared between subjects with comorbidities and those in the non-comorbid groups. Next, we performed ROC analyses of the anti-RBD level at 30%, 60%, and 90% sVNT (% inhibition) levels to determine the best cutoff point with the best accuracy among subjects with no comorbidities. Using these values, we measured the sensitivity, specificity, and positive and negative predictive values for each group of PLWH, SLE, and CKD subjects. The accuracy for each group was finally compared with that for the non-comorbid subjects.

### 2.6. Ethical Clearance

The Health Research Ethics Committee of Universitas Padjadjaran approved this study on 17 May 2021, under ethics number 410/UN6.KEP/EC/2021. The study followed the principles outlined in the Declaration of Helsinki.

## 3. Results

### 3.1. Study Subjects

We enrolled a total of 517 subjects in this study. The subjects consisted of four groups: those with no comorbidities (*n* = 182), PLWH (*n* = 100), SLE (*n* = 92), and CKD (*n* = 143) (Table 1). The PWLH group consisted of subjects who were already on ARV. Most were in stage 1 (*n* = 94), and a small proportion were in stage 2 or higher (*n* = 6). We only had data on CD4 status for 23 PLWH subjects, where 17 subjects had CD4 > 200/mm^3^ and 6 had CD4 < 200/mm^3^.

Based on age, the CKD group had the highest age distribution, while the PLWH group had the lowest age distribution.Gender-wise, the proportion of females was slightly higher than that of males, especially in the SLE group, while the PLWH group was predominantly male. A total of 113 (21.86%) subjects had a history of COVID-19 infection. Among the 517 subjects, 216 subjects (41.8%) were unvaccinated and 301 subjects (58.2%) were vaccinated at the time of the survey (Table 1). Most of the subjects were vaccinated using CoronaVac. At the later stage, when mRNA vaccines were available, some of the non-comorbid subjects and the CKD subjects had received mRNA vaccines.

### 3.2. sVNT Is More Sensitive to Detect Neutralization Capacity than Anti-RBD

Generally, the % inhibition using sVNTs was distributed at a higher level than with the anti-RBD assay (Figure 1a). When using the sVNT, overall, 304 subjects (58.80%) exceeded the middle value of 50% inhibition, while only 214 subjects (41.39%) reached above the middle value of 100 AU/mL in the anti-RBD assay (*p* = 0.0001). We also observed similar distribution patterns in all of the groups (Figure 1a,b).

### 3.3. The Different Levels of Anti-RBD and sVNT in Each Group of Subjects

The subject enrollment time varied widely. There were two survey timepoints for the non-comorbid group: the first and second quarters of 2021, and the third quarter of 2022. The PLWH group was enrolled in the second quarter of 2021. Most SLE and CKD subjects were enrolled in the fourth quarter of 2021. Figure 1a shows that the distribution of anti-RBD titers and sVNT % inhibition among the unvaccinated group were higher in the non-comorbid subjects compared to the PLWH and those with SLE. We think that this may have been due to the comorbid condition in the latter groups. Interestingly, while still unvaccinated, the CKD group showed the highest serological levels. We think that this was due to their frequent visits to the hospital for hemodialysis. The CKD subjects having the highest proportion of history of COVID-19 may also support this hypothesis (Table 1). In the vaccinated subjects, we observed the highest serological levels in members of the non-comorbid group and CKD group who had received mRNA vaccines (Figure 1b). We stratified the respondent in age group and in this observation we found that the older subjects have significantly higher SVNT % inhibition level (Appendix A).

### 3.4. The Effects of Vaccination on the Distribution of Anti-RBD and sVNT Levels

In Figure 2a,b, we can observe that vaccination’s effects were prominent in all groups, as shown by both the anti-RBD and sVNT values. Some non-comorbid subjects, as well as all PLWH and SLE subjects, were vaccinated with CoronaVac. With CoronaVac, we could see a modest increase in the anti-RBD and sVNT levels. The most prominent increases were observed in some non-comorbid and all CKD subjects who had received mRNA vaccines.

### 3.5. Correlation between Anti-RBD Titers and Percentage Inhibition

In Figure 3a, we demonstrate the correlation of anti-RBD titers and the % inhibition measured with the sVNT using a linear Spearman’s ranked correlation test among each group of subjects. We observed a correlation of R = 0.7903 (95% CI; 0.7286–0.8393) among the non-comorbid group. The correlation test for the PLWH group demonstrated R = 0.8877 (95% CI; 0.8372–9.9231) (Figure 3b), which was significantly higher (*p* = 0.0072) compared to the non-comorbid group. The correlation tests for the SLE and CKD groups provided results of R = 0.7843 (95% CI; 0.6904–0.8522) and R = 0.8153 (95% CI; 0.7518–0.8639), respectively. These correlation test results among the SLE and CKD groups were not significantly different from those of the non-comorbid group (Figure 3c,d).

### 3.6. ROC and Cutoff Values to Detect 30%, 60%, and 90% Inhibition of sVNT

We performed ROC analyses to determine the best cutoff of the anti-RBD assay against sVNT inhibition at the 30%, 60%, and 90% levels among the non-comorbid group. At 30% inhibition, we obtained an AUC of 0.955 (95% CI; 0.913, 0.997). Based on the ROC curve, we observed a 28.30 AU/mL anti-RBD value as the best cutoff, with a sensitivity and specificity of 87.9% and 93.0%, respectively (Figure 4a). At 60% sVNT inhibition, we obtained an AUC of 0.946 (95% CI; 0.907, 0.985), with the best cutoff point being an anti-RBD value of 58.48 AU/mL. With this value, we achieved sensitivity and specificity of 91.7% and 90.7%, respectively (Figure 4b). Meanwhile, at 90% sVNT inhibition, we observed an AUC of 0.921 (95% CI; 0.883, 0.960) and obtained the best anti-RBD cutoff value of 95.45 AU/mL. With this value, we obtained sensitivity and specificity of 86.4% and 81.4%, respectively (Figure 4c).

### 3.7. Sensitivity and Sensitivity of Anti-RBD in Various Comorbid Conditions

In Table 2a–c, we present the performance of the anti-RBD test at the 30%, 60%, and 90% sVNT inhibition levels for all of the comorbid groups compared to the non-comorbid group. At the anti-RBD cutoff level of 28.3 AU/mL, we found almost no significant differences in sensitivity and specificity to detect a 30% sVNT inhibition level, except for the SLE group. Significantly lower sensitivity was observed in the SLE group (*p* ≤ 0.0001), while the specificity remained similar. We also observed similar patterns using the anti-RBD cutoff level of 58.8AU/mL to detect 60% sVNT inhibition. Similar sensitivity and specificity were found, except for the SLE group, which showed significantly lower sensitivity (*p* = 0.0003). Meanwhile, at an RBD level of 95.4 AU/mL, there were no significant differences in sensitivity across all comorbid groups to detect 90% sVNT inhibition. However, the specificity was significantly higher in the SLE group (*p* = 0.0076). The differences were mainly observed in the sensitivity of the tests, but specificity remained high. The negative predictive value of the test was shown to be adequately high in all of the groups. This means that there would be a minimal number of cases misclassified as having low serological levels.

## 4. Discussion

We conducted a head-to-head comparison of an anti-SARS-CoV-2-S-RBD antibody test versus the surrogate viral neutralization test in subjects living with HIV, SLE, and CKD. The effects of previous infections and received vaccines were significant and clearly shown. We found that, in general, the results of the anti-RBD assay were well correlated with those of the sVNT in non-comorbid and comorbid subjects. However, we observed a slight change in accuracy. The sensitivity and specificity of the anti-RBD test among PLWH and CKD patients were comparable with those among non-comorbid subjects. However, among subjects with SLE, the anti-RBD assay’s sensitivity to detect 30% and 60% sVNT inhibition was lower, while its specificity to detect 90% sVNT inhibition was higher. However, overall, the specificity and negative predictive value remained at a high level, confirming that this test should still be beneficial to detect low serological levels in comorbid subjects. To the best of our knowledge, this evaluation has not been reported by other researchers.

One method to measure immune response to diseases and vaccines is by measuring specific antibodies or their neutralization capacity [13]. In COVID-19, high anti-RBD or neutralizing antibody titers are linked to a decreased likelihood of symptomatic and severe illness [14]. Point-of-care anti-SARS-CoV-2-RBD tests are available to simplify the method so that it can be readily utilized in resource-limited settings. The correlation of neutralizing activity with anti-SARS-CoV-2-RBD antibody levels is known to be high. However, due to the various methods and reagents used, an absolute conversion factor has not been established [13].

### 4.1. sVNT Is More Sensitive than Anti-RBD

Our study showed that the sVNT results were distributed at a higher range than the anti-RBD results in the same blood samples, meaning that the sVNT is more sensitive to detect the presence of neutralization capacity. This situation may be due to the nature of the test, which identifies total immunodominant neutralizing antibodies that block the interaction between the SARS-CoV-2 surface receptor-binding domain and the angiotensin-converting enzyme 2 (ACE2) receptor protein [14]. The anti-RBD test is less sensitive because it only detects specific antibodies against the human receptor-binding domain (S-RBD) [15]. Anti-RBD test results do not fully represent the whole spectrum of possible neutralizing antibodies. There could be other additional neutralizing antibodies that reinforce the % inhibition detected by the sVNT [13,16].

### 4.2. The Effects of Natural Transmission and Vaccines

Among the unvaccinated subjects, we found significantly higher anti-RBD and sVNT % inhibition levels in the CKD group than in the other groups. This finding could be due to several factors. Firstly, the serosurvey of the CKD subjects was conducted later in the pandemic. The severely impactful SARS-CoV-2 Delta outbreak occurred between July and September 2021 in our area, while the CKD subjects were tested from October to November 2021 [17]. Secondly, the CKD subjects had at least twice-weekly visits to the hospital for their hemodialysis, meaning that they were there more frequently than the other groups. As we also know, COVID-19 is very widely transmitted, especially during outbreaks, and even more so in the hospitals [18,19]. As such, CKD patients may be more exposed to COVID-19 transmission than most other patients registered in hospitals.

We observed an apparent effect of vaccination on the distribution of the anti-RBD and sVNT results. CoronaVac provided a modest increase in antibody levels in all of the groups. The mRNA vaccine showed significantly higher antibody and neutralization levels in the non-comorbid and CKD groups. CoronaVac was delivered in Indonesia starting in March 2021, but the mRNA vaccine was only delivered in August–September 2021. Some non-comorbid subjects received mRNA booster vaccines, while all of the CKD subjects received mRNA vaccines. The modest effects of CoronaVac and higher stimulation effects of mRNA vaccines have been well documented elsewhere [20,21].

### 4.3. Correlation of sVNT vs. Anti-RBD

Our study found that the correlation between the sVNT and point-of-care anti-RBD test is excellent. We found a correlation coefficient of R > 0.7 in all groups. Other reports have shown similar findings. Malipiero et al. tested other point-of-care anti-RBD tests and found correlation coefficients of R = 0.5887 to 0.7332 [13]. However, more sophisticated laboratory methods using ECLIA/ELISA have been known to demonstrate higher correlation coefficients, at 0.8425 to 0.9736 [14]. Interestingly, we found a significantly higher correlation among PLWH. We hypothesized that the sVNT, which utilized all predominant neutralizing antibodies in the reaction, was more reduced than the anti-RBD test, which only measured specific antibodies. Therefore, the lower sVNT results may contribute to the higher correlation coefficients. Other studies have demonstrated that PLWH with lower baseline CD4 counts, lower baseline CD4/CD8 ratios, and greater baseline viral loads had poorer seroconversion rates and immunological titers [22].

### 4.4. Accuracy to Detect Specific sVNT Inhibition Levels

We hypothesized that the comorbid conditions would affect the anti-RBD accuracy. However, we found that, compared to the non-comorbid group, effects on sensitivity and specificity were only found in the SLE group. The sensitivity of the anti-RBD test to detect 30% and 60% sVNT inhibition was lower, and the specificity was higher at 90% inhibition. From other studies, we have learned that COVID-19 vaccination recipients with SLE show a reduced antibody response, even without immune-suppressing medicines. After receiving the COVID-19 immunization, auto-reactive T cells showed decreased activation [23]. On the other hand, immune suppression provided by SLE therapy was mainly directed to decrease the activity of humoral immunity [24]. In this group we observed a predominant lack of humoral immunity, in contrast to the PLWH group, who lacked cellular immunity.

Our observations among CKD subjects did not show any difference in the correlation or the accuracy of the anti-RBD assay compared to non-comorbid subjects. Several studies have pointed out that lower anti-SARS-CoV-2 antibody titers were observed in patients with renal impairment attending hemodialysis [25]. This difference was not seen in our observations, perhaps because most subjects in this group received mRNA vaccines and had maximal anti-RBD and sVNT inhibition levels. Another possibility was that if the comorbid condition similarly reduced both anti-RBD and neutralization levels, there would be no effect on the sensitivity and specificity.

### 4.5. Weaknesses/Limitations

Since the enrollment of subjects was conducted cross-sectionally and at various points in time, our observation may have been affected by the pandemic phases, vaccine types, and the duration between vaccination and our serological assessment. Because of this study design, we could not compare or assess the specific factors within the comorbid group that affected the antibody response. For example, it would be ideal if we could stratify the antibody levels based on the CD4 counts in the PLWH group. With this convenient design, a higher range of antibody titers in some groups who received mRNA vaccines may affect the comparison.

We acknowledge that the reference standard utilized in this study, the sVNT, was not the perfect gold standard. The best standard for the determination of neutralization antibodies is the plaque reduction neutralization test (PRNT). However, to conduct the PRNT requires substantial resources, including a lab with a high level of biosafety. Another alternative is the pseudovirus neutralization test (pVNT), which provides a higher level of correlation to those based on live virus. However, several studies have also highlighted the good correlation between pVNT and sVNT results, as well as the reliability of the sVNT in detecting neutralization antibodies while being much more accessible to most laboratories [26]. A direct comparison of anti-RBD against sVNT may increase the reliability of using point-of-care anti-RBD testing to be utilized in settings with limited resources.

## 5. Conclusions

Compared to the anti-RBD titers, the percentage inhibition values of the sVNT were more sensitive in detecting the presence of neutralizing capacity. The anti-RBD assay had an excellent correlation with the sVNT in the PLWH, SLE, and CKD groups. The higher correlation coefficient among PLWH was hypothesized to be caused by the lower detection capacity of the sVNT. The anti-RBD assay showed high accuracy, with sensitivity > 80% and specificity > 90% in most groups. Lower sensitivity and higher specificity among the subjects with SLE made us consider that this group has the most compromised humoral immunity, related to their illness and immunotherapy. However, overall, the consistent high specificity of the anti-RBD assay in all groups showed that the anti-RBD test can be reliably utilized to detect the low antibody responses or neutralization capacity in subjects with these comorbidities. The low anti-RBD results may suggest the need for additional COVID-19 booster vaccinations.

## Figures and Tables

**Figure 1 vaccines-12-00558-f001:**
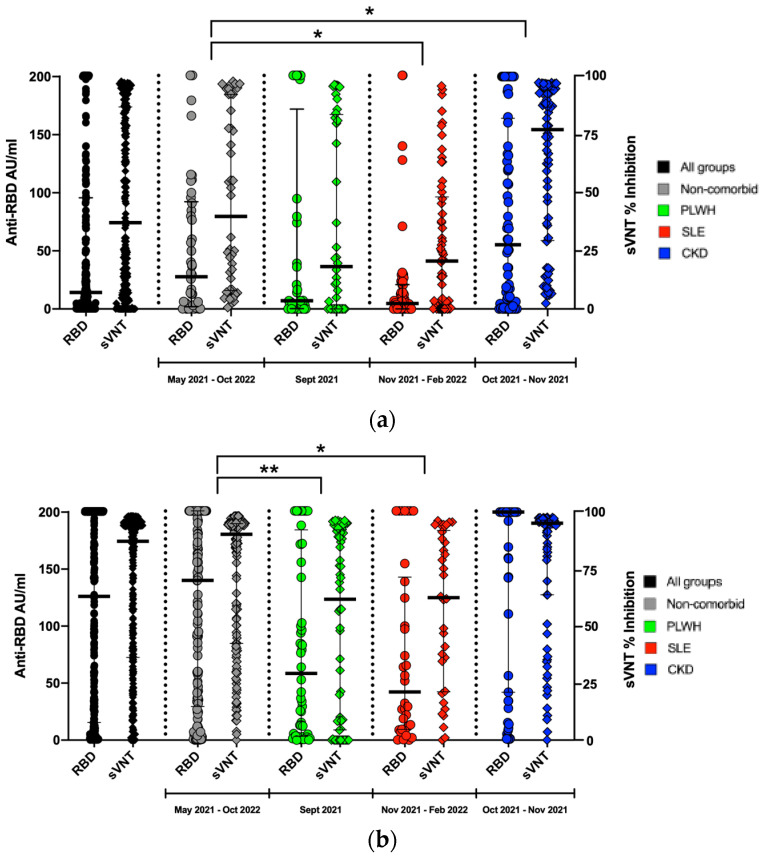
(**a**) Graph plot of anti-RBD and sVNT levels among unvaccinated subjects, stratified by comorbidity classification. (**b**) Graph plot of anti-RBD and sVNT levels among vaccinated subjects, stratified by comorbidity classification. Note * *p* < 0.05, ** *p* < 0.01.

**Figure 2 vaccines-12-00558-f002:**
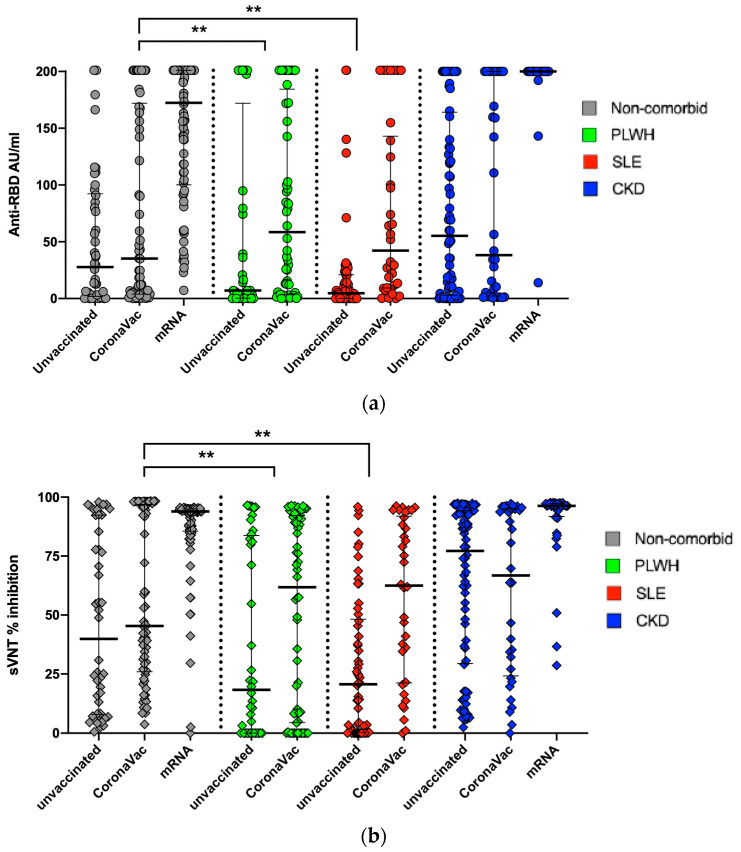
(**a**) Anti-RBD antibody levels among vaccinated and unvaccinated subjects, stratified by comorbidity classification. (**b**) Surrogate viral neutralization % inhibition levels among vaccinated and unvaccinated subjects, stratified by comorbidity classification. Note ** *p* < 0.01.

**Figure 3 vaccines-12-00558-f003:**
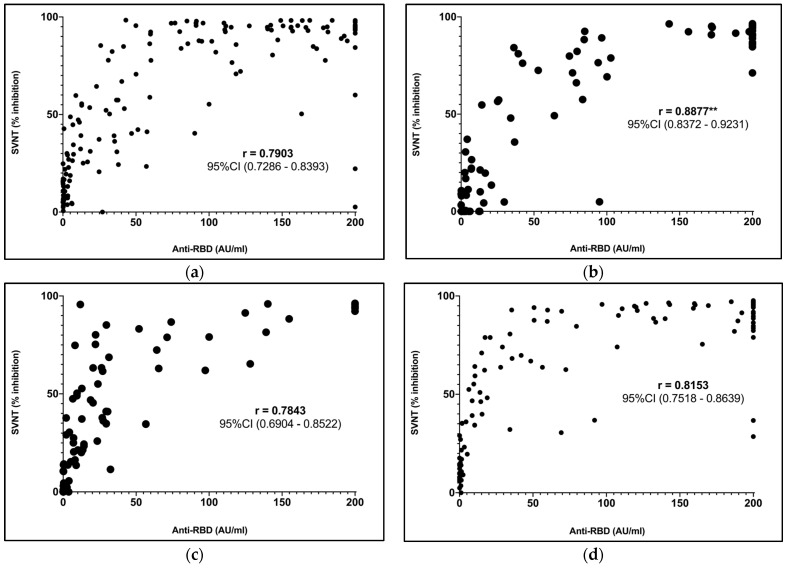
Graph plot and Spearman’s correlation between anti-RBD and sVNT conducted for the (**a**) non-comorbid, (**b**) PLWH, (**c**) SLE, and (**d**) CKD groups.

**Figure 4 vaccines-12-00558-f004:**
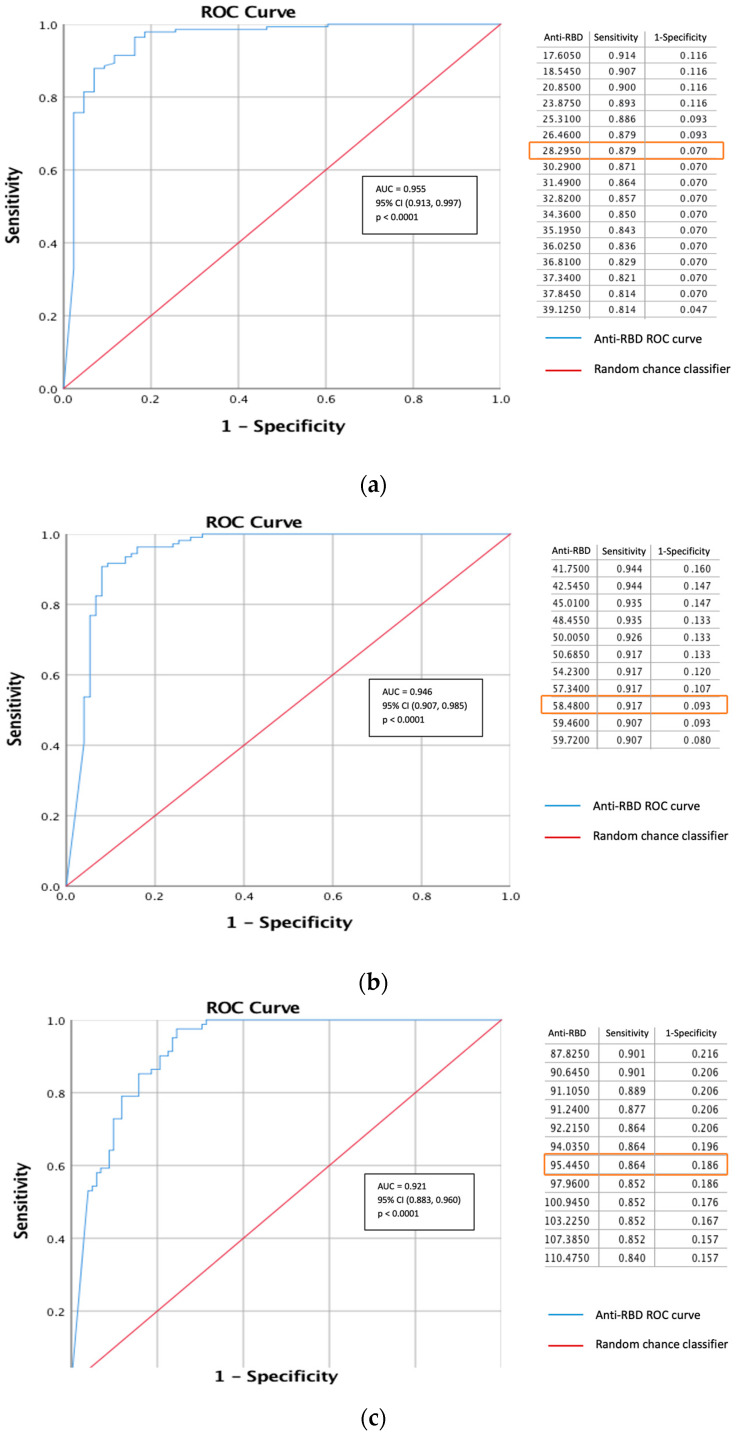
ROC curve of anti-SARS-CoV-2-RBD results at (**a**) 30%, (**b**) 60%, and (**c**) 90% sVNT inhibition levels to determine cutoffs among the non-comorbid subjects. Blue line indicates the ROC curve resulted from the sensitivity/specificity calculation at different cutoff level. Red line indicates a baseline (random) situation if there was no diagnostic value of the observe test.

**Table 1 vaccines-12-00558-t001:** Baseline characteristics of all subjects in the study.

	Non-Comorbid	HIV	SLE	CKD
	N = 182	N = 100	N = 92	N = 143
Age, median (IQR)	40 (30–53)	33.5 (30–41) **	36.5 (27–44) **	48 (40–55) **
Gender, n (%)				
Male	85 (46.70)	85 (85.00) **	4 (4.35) **	65 (45.45)
Female	97 (53.30)	15 (15.00) **	88 (95.65) **	78 (54.55)
History of COVID-19, n (%)				
No	124 (68.13)	86 (86.00) **	83 (90.22) **	111 (77.62) *
Yes	58 (31.87)	14 (14.00) **	9 (9.78) **	32 (22.38) *
Vaccination, n (%)				
Vaccinated	136 (74.73)	60 (60.00) **	34 (36.96) **	71 (49.65) **
Unvaccinated	46 (25.27)	40 (40.00) **	58 (63.04) **	72 (50.35) **
Vaccine Type (%)				
CoronaVac	62 (45.59)	57 (95.00) **	34(100) **	32 (45.07)
mRNA	74 (54.41)	0 (0) **	0 (0) **	39 (54.93)
ChAdOx1-S	0 (0)	3 (5.00) **	0 (0) **	0 (0)
Time Survey Conducted				
2nd Quarter, 2021	54 (29.67)	0 (0)	0 (0)	0 (0)
3rd Quarter, 2021	53 (29.12)	100 (100)	0 (0)	0 (0)
4th Quarter, 2021	1 (0.55)	0 (0)	86 (93.48)	143 (100)
1st Quarter, 2022	0 (0)	0 (0)	6 (6.52)	0 (0)
3rd Quarter, 2022	74 (40.66)	0 (0)	0 (0)	0 (0)
Result				
Anti-RBD (AU),				
Median (IQR)				
Unvaccinated	27.07.00	7	04.08	55.03.00
	(1.9–92.3)	(0.3–172)	(0.1–21)	(3–164.1)
Vaccinated	140.00.00	58.5 *	42.2 *	200 *
	(28.3–201)	(3.3–184.5)	(9.3–201)	(42–200)
sVNT (% inhibition), Median (IQR)				
Unvaccinated	39.09.00	18.03	20.6 *	77.2 *
	(8–92.4)	(0–83.7)	(0.3–48.2)	(29.5–94.3)
Vaccinated	89.59.00	61.8 **	62.05.00	95.01.00
	(42.4–95)	(4.5–92.2)	(21.2–92)	(63.7–96.4)

Significant differences in comparison to the non-comorbid group: * *p* < 0.05; ** *p* < 0.01.

**Table 2 vaccines-12-00558-t002:** (**a**). Accuracy of anti-RBD with a specific cutoff at 30% inhibition of sVNT for each comorbidity group. (**b**). Accuracy of anti-RBD with a specific cutoff at 60% inhibition of sVNT for each comorbidity group. (**c**). Accuracy of anti-RBD with a specific cutoff at 90% inhibition of sVNT for each comorbidity group.

**(a)**					
	**Accuracy to Detect 30% Inhibition Using an Anti-RBD Cutoff of 28.30 AU/mL**				
		**Non-Comorbid**	**HIV**	**SLE**	**CKD**
**Sensitivity**	n/total	121/138	48/53	27/46	97/115
	%	**87.7%**	**90.6%**	**58.7% ****	**84.4%**
	95% CI	(81.0–92.7)	(79.3–96.9)	(43.2–73.0)	(76.4–90.4)
**Specificity**	n/total	40/44	45/47	45/46	27/28
	%	**90.9%**	**95.7%**	**97.8%**	**96.4%**
	95% CI	(78.3–97.5)	(85.5–99.5)	(88.5–99.9)	(81.6–99.9)
**PPV**	n/total	121/125	48/50	27/28	97/98
	%	**96.8%**	**96.0%**	**96.4%**	**99.0%**
	95% CI	(92.2–98.7)	(86.0–98.9)	(79.28–99.5)	(93.39–99.8)
**NPV**	n/total	40/57	45/50	45/64	27/45
	%	**70.3%**	**90.00% ****	**70.31%**	**60.00%**
	95% CI	(59.9–78.8)	(79.6–95.4)	(62.6–77.0)	(49.4–69.8)
**(b)**					
	**Accuracy to Detect 60% Inhibition Using an Anti-RBD Cutoff of 58.48 AU/mL**				
		**Non-Comorbid**	**HIV**	**SLE**	**CKD**
**Sensitivity**	n/total	98/107	40/44	20/30	84/99
	%	**91.6%**	**90.9%**	**66.7% ****	**84.9%**
	95% CI	(84.6–96.1)	(78.3–97.5)	(47.2–82.7)	(76.2–91.3)
**Specificity**	n/total	68/75	53/56	62/62	40/44
	%	**90.7%**	**94.6%**	**100.0%**	**90.9%**
	95% CI	(81.7–96.2)	(85.1–98.9)	(94.2–100.0)	(78.3–97.5)
**PPV**	n/total	98/105	40/43	20/20	84/88
	%	**93.3%**	**93.0%**	**100.0%**	**95.5%**
	95% CI	(87.3–96.6)	(81.5–97.6)	(83.2–100.0)	(89.2–98.2)
**NPV**	n/total	68/77	53/57	62/72	40/55
	%	**88.3%**	**93.0%**	**86.11%**	**72.7% ***
	95% CI	(80.1–93.4)	(83.9–97.1)	(78.9–91.1)	(62.4–81.1)
**(c)**					
	**Accuracy to Detect 90% Inhibition Using an Anti-RBD Cutoff of 95.45 AU/mL**				
		**Non-Comorbid**	**HIV**	**SLE**	**CKD**
**Sensitivity**	n/total	69/80	25/26	11-Dec	65/69
	%	**86.3%**	**96.2%**	**91.7%**	**94.2%**
	95% CI	(76.7–92.9)	(80.3–99.9)	(61.5–99.8)	(85.8–98.4)
**Specificity**	n/total	83/102	66/74	75/80	58/74
	%	**81.4%**	**89.2%**	**93.8% ****	**78.4%**
	95% CI	(72.5–88.4)	(79.8–95.2)	(86.0–97.9)	(67.3–87.1)
**PPV**	n/total	69/88	25/33	Nov-16	65/81
	%	**78.4%**	**75.8%**	**68.8%**	**80.3%**
	95% CI	(70.6–84.6)	(61.8–85.8)	(48.1–83.9)	(72.4–86.3)
**NPV**	n/total	83/94	66/67	75/76	58/62
	%	**88.3%**	**98.5% ****	**98.7% ****	**93.6%**
	95% CI	(77.3–88.6)	(90.6–99.8)	(92.0–99.8)	(84.8–97.4)

Significant difference in comparison to the non-comorbid group. Bold text corresponds to the main calculation result of the sensitivity, specificity, PPV and NPV. * *p* < 0.05; ** *p* < 0.01.

## Data Availability

This article contains the data used to support this study’s findings.

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
