# Peer review of "Accuracy of Anti-SARS-CoV-2 Antibody in Comparison with Surrogate Viral Neutralization Test in Persons Living with HIV, Systemic Lupus Erythematosus, and Chronic Kidney Disease"

_vaccines, 2024, doi:10.3390/vaccines12050558_

Round 1

Reviewer 1 Report

Comments and Suggestions for Authors

The author aims to evaluate the accuracy of anti-RBD detection among different patient cohorts (PLWH, SLE, CKD, without complications). The conclusion drawn is that anti-RBD detection helps in screening populations with low to moderate antibody levels to guide their vaccination against COVID-19. The study is deemed to have certain innovation and clinical application value.

1. The authors should elaborate on the types of antibodies (IgG/IgM) detected using fluorescence immunoassay technology for anti-RBD, as well as the types of antigens targeted (wild-type/mutant S-RBD antigen). The specific type of HRP-RBD antigen used in the SVNT method should also be clarified.

 2. It is suggested that the authors clarify the types of vaccines administered (wild-type vaccine/mutant strain vaccine) and the number of doses among the enrolled volunteers.

 3. A history of COVID-19 infection, as a confounding factor, may affect antibody titers. The proportion of COVID-19 infection is higher in the group without complications, and their COVID-19 antibody levels are higher. This history of COVID-19 infection may influence the results. Additionally, earlier studies have found that PLWH receiving standardized treatment and with CD4>500 have COVID-19 antibody titers similar to non-HIV-infected individuals, while PLWH with lower CD4 counts have lower antibody titers. It is recommended to supplement information on the CD4 status of the PLWH population after treatment. Therefore, the conclusions drawn by the authors regarding the comparison of antibody levels among different patient cohorts need to consider more influencing factors for interpretation.

 4. The results suggest significant differences in the results obtained from the two detection methods for some patients. It is recommended that the authors analyze in detail the general characteristics of these patients and explore possible influencing factors (such as vaccine administration, time since infection, type of infecting virus, patient's underlying conditions, and general information).

 5. Anti-RBD detection assesses the binding ability of antibodies, which is a prerequisite for neutralization ability. It is advisable for the authors to supplement explanations of the methodological comparison between anti-RBD detection and SVNT method detection. Moreover, as SVNT serves as an alternative experiment for pseudovirus neutralization, the authors should supplement information on the feasibility of using SVNT as a reference method.

 6. Current research suggests that vaccination with the wild-type COVID-19 vaccine has limited efficacy against subsequent infection with mutant viruses. If the method detects antibodies against the wild-type COVID-19 virus, it may not accurately measure the body's protection against mutant strains.

 7. Figure 1a mentioned in the text is not labeled in the figure.

 8. Statistical methods for Table 1 are not listed.

 9. Sections 3.4 and 3.5 in the results section are repetitive and need to be corrected.

Comments on the Quality of English Language

Minor editing of English language required.

Author Response

Dear Reviewer, thank you for your evaluation. Please see the attachment containing the item by item responses to your evaluation.

Reviewer 2 Report

Comments and Suggestions for Authors

Title: Accuracy of Anti-SARS-CoV2 Antibody in Comparison with Surrogate Viral Neutralization Test in Person Living with HIV, Systemic Lupus Erythematosus and Chronic Kidney Disease

-- Summary of the main findings of the study.

The study assessed the accuracy of the anti-RBD test compared to sVNT as a reference in people with co-morbidity: people living with HIV (PLWH), systemic lupus erythematosus (SLE), and chronic kidney disease (CDK). Enrolment included 182 non-comorbid subjects and 335 patients with co-morbidities.

Strengths: The article is well structured and generated a conclusion supported by the results.

Limitations: Limited analyses. Discuss other stats metrics.

Comments:

       Can the subjects who received Sinovac followed by mRNA vaccines be incorporated in Table 1 and Figure 2?

       Are either test known to be performing differently with different variants of COVID-19?

       Figure 1 has an incomplete legend. 1a part is missing.

       Paragraph 3.4 is a duplicate of 3.5.

       Fix labeling in Figure 2. Inconsistent labeling on the x-axis between 2a and 2b.

       Figure 2b legend is missing the words “vaccinated and”. The sentence should read: “Surrogate viral neutralization % inhibition level among vaccinated and unvaccinated subjects stratified by comorbidity classification”.

       Line 194, what is CoronaVac referring to? It is not defined in Methods.

       Figure 4, add header label for the 3 columns on the right side of ROC graphs.

       Line 329-331, incomplete sentence, missing what it is compared to.

       There are notably better PPV at levels of anti-RBD corresponding to 30% and 60% inhibition – PPV was even greater than NPV for these levels. Discuss the clinical implication and/or application and what this mean in terms of the optimal anti-RBD cut-off (AU/mL).

Comments on the Quality of English Language

There are a few syntax and missing sentence issues.

Author Response

Dear Reviewer, thank you for your evaluation, attached below we send the item by item responses to your evaluation in a word document. 

Reviewer 3 Report

Comments and Suggestions for Authors

This manuscript compares and evaluates the point-of-care antibody test with the surrogate viral neutralisation test (sVNT) as a reference standard. The writing and methodology were well-written. The outcomes are also well-described and clear.

Major concerns.

1. A subgrouping of these participants by disease (non-comorbid, PLWH, SLE and CKD) may differ by age. Almost all CKD are middle-aged and elderly. In contrast, SLE and recent HIV infections occur in younger.

Thus, subgrouping by age group [Children-Adolescence (if available), young adult, middle-aged, and elderly] should be suggested to clarify in Table 1.
This data may discuss immunogenicity by age-dependent if your outcomes work.

Minor concerns.

1. The p-value: In 3.8, the p-value was six decimals (e.g. 0.000014), over precision compared to the small sample size in subgrouping (n=1xx).

Suggest using four decimals (e.g. 0.0072, <0.0001) to make it consistent with the p-value in the other section.

Comments.

1. Suggest using "sVNT" instead of "SVNT" to make it consistent with almost all forms in the literature.

2. Suggest using a research or commercial name for COVID-19 vaccines instead of a manufacturer name. The reason is that the same manufacturer may produce vaccines in more than 1 type, such as bivalent, lite version, XBB monovalent, etc. Using only the manufacturer's name may lead to confusion about which vaccines are used in this study.

- Commercial name; CoronaVac, Covilo, Vaxivria, Comirnaty, Spikevax.
- Research name; CoronaVac, BBIBP-CorV, ChAdOx1-S, BNT162b2, mRNA-1273.

Typos.

1. Line 41: "Coronavirus disease 2019 (COVID-19)"

2. Line 93: "in he mRNA group."
Was "in the..."?

3. Lines 161, 254, 257, 260 and 315: Suggest using a dot " . " as a decimal separator to make it consistent with all English styles, not French or other styles.

4. Line 287: "SARS-CoV2".

Author Response

Dear Reviewer, thank you for your evaluation, attached below we send the item by item responses to your evaluation in word document. 
